# Development and Evaluation of Liposomal *Celastrol*-PROTACs for Treating Triple-Negative Breast Cancer

**DOI:** 10.3390/ph18091381

**Published:** 2025-09-16

**Authors:** Xuebin Li, Chaoqun Yu, Gongyi Zheng, Yanghong Li, Weiguo Cao, Fan Wang

**Affiliations:** 1College of Chinese Materia Medica, Chongqing University of Chinese Medicine, Chongqing 402760, China; 2College of Traditional Chinese Medicine, Chongqing Medical University, Chongqing 400016, China; 15895907621@163.com (X.L.); m18869522959@163.com (G.Z.);; 3Research Center of Pharmaceutical Preparations and Nanomedicine, College of Pharmacy, Chongqing Medical University, Chongqing 400016, China; 203241@cqmu.edu.cn

**Keywords:** triple-negative breast cancer, *Celastrol*-PROTACs, PEGylated liposomes, targeting efficiency, reduce toxicity

## Abstract

**Background:** Based on our previous study, *Celastrol*-based proteolysis-targeting chimeras (*Celastrol*-PROTACs) were shown to induce apoptosis in 4T1 cells by selectively degrading GRP94 and CDK1/4 through the endogenous ubiquitin–proteasome system. However, their clinical translation is limited by poor solubility, low targeting efficiency, and liver and kidney toxicity. **Methods:** To address these limitations, we developed a pegylated liposomal formulation of *Celastrol*-PROTACs (Lip-*Celastrol*-PROTACs) and evaluated its therapeutic efficacy and safety profile. **Results:** The tumor volume of the mice in the *Celastrol*-PROTACs solution group (286 ± 79 mm^3^) was significantly larger than that of those in the Lip-*Celastrol*-PROTACs group (229 ± 49 mm^3^) on day 18 after intravenous administration (*p* < 0.01). This difference between the two groups was statistically significant (*p* < 0.01). Notably, the *Celastrol*-PROTACs group exhibited significantly greater weight loss compared to the Lip-*Celastrol*-PROTACs group (*p* < 0.001). In vivo toxicity assessments revealed that the levels of AST and BUN in the *Celastrol*-PROTACs group were 27.93 ± 4.88 U/L and 12.36 ± 1.33 μmol/L, respectively, whereas those in the Lip-*Celastrol*-PROTACs group were found to be 7.92 ± 0.94 U/L and 8.19 ± 0.67 μmol/L, respectively. These findings indicate a statistically significant difference between the two formulations (*p* < 0.01). **Conclusions:** Our research demonstrated that pegylated liposomes could improve the targeting efficiency and minimize the toxicity of PROTACs, thereby improving overall therapeutic efficacy. These findings indicated that Lip-*Celastrol*-PROTACs represent a promising strategy for future clinical applications.

## 1. Introduction

Breast cancer, a malignant tumor that seriously threatens the health of women worldwide, has now surpassed lung cancer to become the cancer with the highest number of diagnosed cases [1]. Among the various subtypes of breast cancer, triple-negative breast cancer (TNBC) is characterized by the absence of estrogen receptor (ER), progesterone receptor (PR), and human epidermal growth factor receptor 2 (HER2). This subtype accounts for approximately 15–20% of all breast cancer cases [2]. Compared with other breast cancer subtypes, TNBC exhibits higher invasiveness and metastatic potential [3,4]. According to current clinical data on TNBC treatment [5,6,7], the median overall survival period for patients is 10.2 months and the 5-year survival rate for patients with localized tumors is 65%, while the 5-year survival rate for patients with distant organ metastasis is only 11%. *Celastrol* has been demonstrated to induce apoptosis in breast cancer cells via the activation of the endoplasmic reticulum stress-mediated PERK signaling pathway [8]. However, its clinical application is constrained by several factors, including poor water solubility, low bioavailability, and the potential for adverse effects such as abdominal pain, dizziness, and nephrotoxicity [9,10]. Moreover, early-stage TNBC is often difficult to diagnose, and the advancement of targeted therapies lags behind that of other subtypes, leading to limited therapeutic options [11]. Therefore, in the clinical management of TNBC, there is an urgent need for novel and highly efficacious therapeutic agents to address this challenge.

Protein-targeting chimeras (PROTACs) demonstrate distinct advantages over existing chemotherapeutic agents, attracting significant attention and research in the field of tumor studies [12,13,14,15]. PROTACs represent an innovative paradigm in drug design that harnesses the degradation mechanisms of the ubiquitin–proteasome system within cellular environments. This approach enhances drug efficacy while mitigating toxicity and partially circumventing drug resistance. In our previous study, we synthesized a *Celastrol*-thalidomide chimera (*Celastrol*-PROTACs), wherein the natural compound *Celastrol* recognizes the target protein, while thalidomide recruits and binds to the E3 ubiquitin ligase, thereby inducing ubiquitination and subsequent degradation of the target protein for therapeutic purposes. Further investigations revealed that the synthesized *Celastrol*-PROTACs effectively degraded GRP94, cyclin-dependent kinase 1 (CDK1), and CDK4, inducing apoptosis and cell cycle arrest in 4T1 cells. Additionally, *Celastrol*-PROTACs exhibited superior tumor inhibition effects in 4T1 tumor-bearing mice compared to *Celastrol* or thalidomide alone.

*Celastrol*-PROTACs display significant therapeutic advantages for TNBC; however, they encounter challenges and limitations in practical applications. In initial investigations, the synthesized *Celastrol*-PROTACs exhibited inadequate aqueous solubility and tissue permeability due to their high molecular weight, which impeded drug distribution and absorption in vivo and consequently diminished therapeutic efficacy [16,17]. Moreover, dimethyl sulfoxide (DMSO), which was used as a solvent in previous studies, is not suitable for clinical application owing to its intrinsic biological toxicity. To address the issues of low targeting specificity and poor solubility associated with these prodrugs, there is an urgent need to design innovative formulations.

Nanotechnology can effectively enhance targeted drug delivery, and not only improves therapeutic efficacy but also mitigates adverse reactions in non-targeted tissues [18,19]. Nano-PROTACs incorporate PROTAC molecules into the core or surface of nanodrug delivery systems through chemical conjugation, physical encapsulation, or electrostatic adsorption. This integration enhances the solubility, permeability, and targeting capabilities of PROTAC molecules, thereby improving their in vivo distribution and metabolic profiles [4,10,17]. Liposomes, with their high biocompatibility, are considered one of the most successful nanomedicine delivery systems for clinical application [20]. For poorly soluble drugs, liposomes can encapsulate them within a phospholipid bilayer, eliminating the need for organic solvents such as DMSO, thereby reducing formulation toxicity. Furthermore, liposomes possess the capability to target tumors effectively and facilitate sustained drug release, thereby enhancing therapeutic efficacy while reducing systemic toxicity [21,22]. Consequently, in this study, liposomes were employed to encapsulate the *Celastrol*-PROTACs prepared in the prior study, aiming to solve the problems of poor solubility and low targeting efficiency of PROTACs.

In this study, polyethylene glycol (PEG) was chosen as one of the key materials for preparing liposomes to obtain spatially stable liposomes of *Celastrol*-PROTACs. These liposomes could improve the tumor targeting specificity of the drug and thus enhance the anti-tumor efficacy. The efficacy and toxicity of the prepared liposomes were comprehensively evaluated both at the cellular level and in animal models. The results demonstrated that the liposomes exhibited significantly superior therapeutic effects and mitigated adverse reactions compared with free *Celastrol*-PROTACs in solution.

## 2. Results and Discussion

### 2.1. Characterization of Lip-Celastrol-PROTACs

The Lip-*Celastrol*-PROTACs prepared via the thin-film hydration method appeared translucent pale yellow under visual inspection (Figure 1A). Transmission electron microscopy (TEM) analysis demonstrated a spherical morphology with an average diameter of approximately 85 nm, showing uniform size distribution and no significant aggregation (Figure 1B). Dynamic light scattering (DLS) analysis demonstrated that Lip-*Celastrol*-PROTACs exhibited a mean particle size of 65.85 ± 0.56 nm with a polydispersity index (PDI) of 0.351 ± 0.0009 (Figure 1C) and a zeta potential of −6.77 ± 0.82 mV (Figure 1D). The literature [23,24] indicates that the particle size of liposomes, if excessively large (>300 nm) or excessively small (<10 nm), can negatively impact drug targeting. Larger particles tend to accumulate excessively at non-targeted sites [25,26], reducing drug specificity and enhancing toxic side effects. Conversely, very small particles (<10 nm) are prone to rapid removal via kidney filtration upon entering systemic circulation [27]. When liposome drug particles are within the range of 50–150 nm, they are less likely to be detected and captured by the endothelial network system, allowing them to remain stable in the bloodstream for extended periods. This enables tumor targeting through the EPR effect [28,29].

The encapsulation efficiency (EE) and drug loading (DL) of Lip-*Celastrol*-PROTACs, measured by HPLC, reached 95.73 ± 4.53% and 10.57 ± 2.96%, respectively (Table 1), indicating high encapsulation performance. *Celastrol*-PROTACs are highly hydrophobic drugs. The hydrophobic groups in *Celastrol*-PROTACs have a strong affinity with the hydrophobic groups of phospholipids and cholesterol. This also enables more *Celastrol*-PROTACs molecules to be loaded onto the bilayer membrane composed of cholesterol and phospholipids in the liposomes. In addition, studies have shown [30] that cholesterol can improve the fluidity of lipid membranes and thus improve the encapsulation efficiency of drugs. The cholesterol content of Lip-*Celastrol*-PROTACs is 10%, which can effectively avoid the leakage of drugs in the liposomes, thereby obtaining a high encapsulation efficiency for *Celastrol*-PROTACs.

### 2.2. In Vitro Hemolysis Safety

Phospholipids are susceptible to hydrolysis during production and storage, with lysophospholipids being one of the primary degradation products. Elevated concentrations of lysophospholipids can cause cell membrane disruption, leading to hemolysis or cellular necrosis [31,32].

In vitro hemolysis assays were performed to assess the hemocompatibility of Lip-*Celastrol*-PROTACs. The colorless supernatant observed after incubation with a 2% (*w*/*v*) erythrocyte suspension confirmed negligible hemolysis (Figure 2A). At concentrations below 100 μg/mL, the hemolysis rate remained under 5%, complying with pharmacopeial safety thresholds. Lip-*Celastrol*-PROTACs exhibited negligible hemolytic activity and vascular irritation at concentrations ≤100 μg/mL, demonstrating excellent hemocompatibility and suitability for injectable administration.

### 2.3. Cellular Uptake

Rhodamine B (RhB)-labeled liposomes (RhB-Lips) were selected to evaluate cellular uptake efficiency. In Figure 3, the red fluorescence represents RhB-Lips, and the blue fluorescence indicates the cell nuclei stained with DAPI. The results showed that after the drugs and cells were co-incubated for 1 h, signs of tumor cellular uptake of the liposomes began to appear. As the incubation time increased, the fluorescence intensity within the tumor cells gradually increased, indicating that the liposomes could be effectively taken up by the tumor cells.

### 2.4. In Vitro Cytotoxicity

The cytotoxicity of the *Celastrol*-PROTACs solution and Lip-*Celastrol*-PROTACs against 4T1 cells was evaluated using the CCK-8 assay. Cells were treated with varying concentrations of *Celastrol*-PROTACs solution or Lip-*Celastrol*-PROTACs, and cell viability was monitored at different time points. Both formulations exhibited concentration- and time-dependent cytotoxic effects on 4T1 cells (Figure 4A–D). Lip-*Celastrol*-PROTACs significantly reduced cell viability compared to the *Celastrol*-PROTACs solution at 2.5 and 3 μg/mL after 6 h (*p* < 0.05) (Figure 4A–D). This reduction persisted at 8 h (2.0, 2.5, and 3 μg/mL; *p* < 0.05) (Figure 4A–D) and became more pronounced at 12 and 24 h across all tested concentrations (0.5–3 μg/mL, *p* < 0.05) (Figure 4A–D). The results demonstrated that, compared with the *Celastrol*-PROTACs solution, Lip-*Celastrol*-PROTACs exhibited significantly enhanced toxicity to tumor cells as the concentration increased and the incubation time was extended.

The IC50s of *Celastrol*-PROTACs solution and Lip-*Celastrol*-PROTACs were further calculated. After 24 h, the IC50 values for 4T1 cells were 1.770 µM and 1.257 µM, respectively, confirming the superior cytotoxicity of the liposomal formulation (Table 2). The toxicity of the drug-loaded liposomes towards tumor cells was significantly higher than that of the drug solution. It is speculated that this is caused by the difference in the drug absorption mechanism at the cellular level for the two formulations. For the *Celastrol*-PROTACs solution, the drugs exist as free molecules and exert their effects through passive diffusion into tumor cells. By contrast, the phospholipid bilayer of Lip-*Celastrol*-PROTACs exhibits excellent compatibility with the cell membrane, enabling drug delivery via adsorption-mediated lipid exchange or direct internalization through endocytosis or membrane fusion. The aforementioned cellular uptake studies also demonstrated that Lip-*Celastrol*-PROTACs can be rapidly and efficiently taken up into cells, leading to reduced cell survival in breast cancer 4T1 cells compared to the *Celastrol*-PROTACs solution.

### 2.5. Wound Healing Assay

Metastatic susceptibility is a hallmark of TNBC. To evaluate the inhibitory effect of Lip-*Celastrol*-PROTACs on cell migration, a wound healing assay was conducted. In the control group (blank culture medium), 4T1 cells nearly repopulated the wound area, whereas Lip-*Celastrol*-PROTACs significantly suppressed migration in a concentration-dependent manner (Figure 5A,B). Near-complete inhibition of cell migration was achieved at 400 nM and 500 nM, demonstrating potent anti-migratory activity. The results demonstrated that Lip-*Celastrol*-PROTACs exerted an obvious inhibitory effect on the proliferation and migration of breast cancer cells.

### 2.6. Cell Viability Analysis

The 4T1 cells were stained with calcein AM and propidium iodide (PI) after 24 h of incubation with blank liposomes, *Celastrol*-PROTACs solution, and Lip-*Celastrol*-PROTACs, respectively. No significant cell death was observed in either the blank control group or the group treated with 250 nM *Celastrol*-PROTACs solution (Figure 6). However, treatment with 250 nM and 500 nM Lip-*Celastrol*-PROTACs resulted in a marked reduction in viable cell counts (Figure 6). These findings indicate that the liposomal formulation exhibited enhanced tumoricidal activity compared to the free drug in solution.

### 2.7. In Vivo Toxicity

Mice were injected intravenously with the *Celastrol*-PROTACs solution and Lip-*Celastrol*-PROTACs at a dose of 6 mg/Kg. The injections were given once every other day for a total of three times. After the mice were sacrificed, their major organs (heart, liver, spleen, lung, kidney) and blood were collected for H&E staining and hematological examination. Histopathological examination (Figure 7A) revealed no significant abnormalities or organ damage in mice treated with Lip-*Celastrol*-PROTACs. However, the *Celastrol*-PROTACs solution group exhibited minimal inflammatory cell infiltration in the periportal regions of liver tissue and localized atelectasis in lung tissue. Occasional inflammatory cell infiltration was noted in certain peribronchial and alveolar interstitial areas, accompanied by thickening of some alveolar walls and evident capillary congestion within the alveolar walls.

Quantitative serum biochemical analysis of alanine aminotransferase (ALT), aspartate aminotransferase (AST), creatinine (CRE), and urea (UA) demonstrated that AST and UA levels in the *Celastrol*-PROTACs solution group were significantly higher than those in the Lip-*Celastrol*-PROTACs group (*p* < 0.01), indicating reduced hepatorenal toxicity of the liposomal formulation. In this paper, liposomes were chosen as drug carriers for *Celastrol*-PROTACs, avoiding usage of toxic solvents such as DMSO. The excellent biocompatibility can effectively reduce the toxic side effects of *Celastrol*-PROTACs and enhance their potential for clinical application.

### 2.8. In Vivo Biodistribution

The biodistribution of liposomes in 4T1 tumor-bearing female BALB/c mice is illustrated in Figure 8. Fluorescence imaging revealed that free DiR (Free-DiR) was predominantly distributed in the left thoracic cavity and abdomen at 2 h post-injection, with no significant accumulation observed at the tumor site. Its fluorescence intensity diminished and disappeared by 72 h. In contrast, DiR-labeled liposomes (DiR-LPs) rapidly accumulated in the tumor region within 2 h. Over time, the fluorescence signal in the tumor area gradually intensified, peaking at 24 h and maintaining strong signals for over 120 h.

Liposomes in this study were formulated using DSPE-mPEG2000. The “stealth effect” of PEGylation minimized recognition by the reticuloendothelial system (RES), enabling prolonged circulation stability. Furthermore, liposomes with particle sizes within 50–150 nm can achieve targeted tumor accumulation via the enhanced permeability and retention (EPR) effect [33,34,35].

### 2.9. Anti-Tumor Efficacy In Vivo

To evaluate the anti-tumor efficacy of Lip-*Celastrol*-PROTACs in vivo, a tumor-bearing BALB/c mouse model was established using 4T1 breast cancer cells. Tumor-bearing mice were randomly divided into five groups. The treatment protocol involved administering the drug every other day for a total duration of 12 days. As illustrated in Figure 9C, tumors in the control group exhibited rapid growth, achieving a final volume of 421 ± 110 mm^3^ after the treatment period. Both the *Celastrol*-PROTACs solution and Lip-*Celastrol*-PROTACs significantly inhibited tumor progression. On day 18, the tumor volumes were 286 ± 79 mm^3^ for the *Celastrol*-PROTACs solution group and 229 ± 49 mm^3^ for the Lip-*Celastrol*-PROTACs group, with statistically significant differences between the two groups (*p* < 0.01) (Figure 9C). After treatment, tumors were resected and weighed. The Lip-*Celastrol*-PROTACs group significantly reduced tumor weights (*p* < 0.05) and enhanced tumor inhibition rates (*p* < 0.05) compared with those of the *Celastrol*-PROTACs solution group (Figure 9D,E). The results demonstrated that Lip-*Celastrol*-PROTACs exhibited superior anti-tumor effects compared with the *Celastrol*-PROTACs solution. Based on previous studies, it could be seen that Lip-*Celastrol*-PROTACs might enhance the toxicity of the drug against tumor cells, increase the tumor cellular uptake rate, and improve the targeted distribution of the drug at the tumor site. Therefore, pharmacological studies here showed better therapeutic effects.

Histopathological analysis of tumor sections was employed to further validate the antitumor efficacy of Lip-*Celastrol*-PROTACs. The tumor cells in the PBS group had normal morphology, were closely arranged, and had abundant angiogenesis. The local vascular density was high, and the vessel walls were dilated. In contrast, tumor tissue cells treated with *Celastrol*-PROTACs solution and Lip-*Celastrol*-PROTACs exhibited an enlarged intercellular gap. In these two groups, tumor cells were relatively loosely arranged, displayed atypia, and showed necrosis in both superficial and deep layers. Notably, the reduction in tumor cells caused by Lip-*Celastrol*-PROTACs treatment was more significant, and the tumor structure suffered even more severe damage. These findings demonstrated that Lip-*Celastrol*-PROTACs possess superior antitumor activity compared with the *Celastrol*-PROTACs solution.

Body weight changes, an important parameter for reflecting systemic toxicity, were detected in this study (Figure 9B). The weight changes of mice treated with Lip-*Celastrol*-PROTACs were not significant and were comparable to those of the control group treated with PBS. However, the *Celastrol*-PROTACs solution group showed substantial weight loss (*p* < 0.001), indicating that the liposome carrier could reduce the systemic toxicity of Celastrol-PROTACs.

### 2.10. Conclusions

In this research, the developed Lip-*Celastrol*-PROTACs, characterized by a particle size of 65.85 ± 0.56 nm and high encapsulation efficiency of 95.73 ± 4.53%, showed enhanced cytotoxic effects compared to the *Celastrol*-PROTACs solution in the 4T1 cell line. The in vivo distribution analysis indicated that the liposomes could effectively accumulate at the tumor site. The in vitro cell uptake study confirmed that the liposomes could be successfully taken up by 4T1 cells. Results from the in vivo pharmacodynamic experiments revealed that Lip-*Celastrol*-PROTACs could significantly suppress tumor growth and provide superior therapeutic outcomes compared to the *Celastrol*-PROTACs solution. The in vivo toxicity assessment demonstrated that Lip-*Celastrol*-PROTACs could markedly reduce the adverse effects of the drug on liver and kidney function while partially mitigating its impact on body weight in tumor-bearing mice. In summary, Lip-*Celastrol*-PROTACs could eliminate the need for toxic solvents, improve tumor targeting, enhance treatment effectiveness, and decreased drug-related toxicity, showing promising potential for clinical applications.

## 3. Materials and Methods

### 3.1. Materials

Soybean phosphatidylcholine (SPC), N-(Carbonyl-methoxypolyethylene glycol 2000) 1,2-distearoyl-sn-glycerol-3-phosphoethanolamine (DSPE-mPEG_2000_) and cholesterol were obtained from AVT Pharmaceutical Technology C, Ltd. (Shanghai, China). Calcein AM/PI double stain kit, Catlog: BB-4126, Bestbio (Shanghai, China). Roswell Park Memorial Institute (RPMI) 1640 medium, and fetal bovine serum (FBS) were purchased from VivaCell Industries (Shanghai, China).

The murine breast cancer cell line 4T1 (4T1 cells) was acquired from the Type Culture Collection of the Chinese Academy of Sciences in Shanghai, China. Healthy male ICR mice, weighing between 18 and 20 g, were procured from the Laboratory Animal Center at Chongqing Medical University in Chongqing, China.

All animal procedures conducted in this study were carried out in accordance with the Guidelines for the Care and Use of Laboratory Animals established by Chongqing Medical University. These procedures received approval from the Animal Ethics Committee of Chongqing Medical University (SCXK2018-0003).

### 3.2. Methods

#### Validation of HPLC Method for Celastrol-PROTACs Determination 

The concentration of *Celastrol*-PROTACs was determined using high-performance liquid chromatography (HPLC, LC-20AD, Shimadzu, Japan) equipped with a Wondasil C18 phase column (4.6 × 250 mm, 5 μm particle size). The mobile phase was a mixture of methanol and 0.1% phosphate buffer (87:13, *v*/*v*). A sample volume of 10 µL was analyzed using a UV detector at a temperature of 35 °C and a wavelength of 450 nm. A standard curve based on peak area (A, mAUs) and *Celastrol*-PROTACs concentration (C, 2.5–100 µg/mL) was drawn, and the equation obtained from linear regression analysis was A = 2637 + 181.66 (R^2^ = 0.9997). The repeatability test data for the *Celastrol*-PROTACS analytical method demonstrated a relative standard deviation (RSD) of 0.78%, indicating satisfactory repeatability of the assay. Furthermore, the RSD values obtained from three sets of sample solutions were all below 2.00%, suggesting that the analytical method exhibits good precision.

### 3.3. Preparation of Lip-Celastrol-PROTACs

The thin-film evaporation method was used for preparing Lip-*Celastrol*-PROTACs. Briefly, 1 mg of *Celastrol*-PROTACs, 34 mg of SPC, 1 mg of DSPE-mPEG_2000_, and 4 mg of cholesterol were accurately weighed and mixed, followed by dissolution in a dichloromethane-based organic solvent mixture. Dichloromethane was removed under vacuum using a rotary evaporator with a water bath temperature of 50 °C, resulting in the formation of a thin lipid film. An appropriate volume of ultrapure water was subsequently added to the flask, and the lipid film was hydrated by shaking at 180 rpm at 37 °C. Finally, the hydrated liposomes were subjected to probe sonication at 150 W for 5 min to yield the final formulation.

### 3.4. Characterization of Lip-Celastrol-PROTACs

The mean particle size and polydispersity index (PDI) of Lip-*Celastrol*-PROTACs were determined by dynamic light scattering (DLS). The liposomes were diluted with ultrapure water (1:25, *v*/*v*) to obtain measurement samples. Particle size and zeta potential (mV) were measured using a Malvern Zetasizer^®^ (ZEN3600, Malvern Instruments, Dumfries, UK) at an instrument temperature of 25 °C and a scattering angle of 173°.

Morphological analysis of Lip-*Celastrol*-PROTACs was performed using a transmission electron microscope (TEM, JEOL JEM-2100, Tokyo, Japan). Samples diluted using ultrapure water were placed on a 200-mesh carbon membrane grid. Subsequently, the samples were negatively stained with phosphotungstic acid. After air drying at 25 °C, the copper mesh was transferred to a microscope sample holder for imaging.

### 3.5. Encapsulation Efficiency and Drug Loading

The encapsulation efficiency (EE) and drug loading (DL) of Lip-*Celastrol*-PROTACs were measured using an ultrafiltration method. Briefly, liposomes were centrifuged at 12,000 rpm at 4 °C for 15 min using an ultrafiltration centrifugal tube (MWCO = 8–12 kDa). After centrifugation, the transparent solution containing the unencapsulated drug was separated. To determine the total amount of *Celastrol*-PROTACs in liposomes, Lip-*Celastrol*-PROTACs were diluted 10-fold in methanol to destroy the liposome structure. Finally, quantitative analysis of *Celastrol*-PROTACs in each sample was performed using the above HPLC method. To calculate the EE and DL, the two equations presented below were used:EE (%) = (M_T_ − M_U_)/M_T_ × 100%(1)DL (%) = (M_T_ − M_U_)/(M_T_ + M_L_) × 100%(2)
where M_T_ was the total amount of *Celastrol*-PROTACs in suspension, M_U_ was the amount of unencapsulated *Celastrol*-PROTACs in liposomes, and M_L_ was the total mass of lipids including SPC, DSPE-mPEG_2000_, and cholesterol.

### 3.6. In Vitro Hemolysis Safety

An in vitro hemolysis experiment was used to analyze the safety of Lip-*Celastrol*-PROTACs for injection in vivo. Serum was extracted from normal mouse blood using a refrigerated centrifuge(Haier, Chongqin, China). An appropriate volume of phosphate buffer was added to the resulting blood, which was then washed two to three times and subsequently centrifuged at 1500 rpm for 15 min until the supernatant appeared colorless. A suspension of red blood cells at a concentration of 2% (*w*/*v*) was prepared using the resuspension method and incubated with varying concentrations of Lip-*Celastrol*-PROTACs for 4 h at 37 °C. Concurrently, negative and positive controls were treated with phosphate buffer and ultrapure water, respectively.

To quantitatively assess hemolysis, samples collected at the end of the incubation period were centrifuged at 2200 rpm for 10 min. One hundred microliters (μL) of the supernatant was carefully transferred into a 96-well plate, and hemoglobin concentration was determined by measuring absorbance at 550 nm using a microplate reader (Thermo Fisher Scientific LabServ^®^, Shanghai China). The hemolysis rate (HR%) was calculated according to the following formula:HR (%) = (A_t_ − A_n_)/(A_p_ − A_n_) × 100%(3)
where A_t_, A_n_ and A_p_ represent the absorbance of the test group, negative control group, and positive control group, respectively.

### 3.7. Cell Culture

4T1 cells were cultured in RPMI-1640 (Gibco, Shanghai, China). All culture media were supplemented with 10% (*v*/*v*) fetal bovine serum (FBS, VivaCell, Shanghai, China) and 1% (*v*/*v*) penicillin–streptomycin (Beyotime, Shanghai, China). The cells were grown at 37 °C in a humidified incubator containing 5% CO_2_. Unless otherwise noted, all in vitro experiments used 4T1 cells.

### 3.8. Cellular Uptake

Since *Celastrol*-PROTACs were non-fluorescent, cell uptake was carried out using RhB-loaded liposomes (RhB-LPs). The preparation of RhB-LPs was the same as that of Lip-*Celastrol*-PROTACs, with a molar ratio of 1:500 between fluorescent dyes and total lipids. Confocal laser scanning microscopy (CLSM, FV1200, Olympus, Tokyo, Japan) was utilized for qualitative evaluation and photography of cellular uptake. Briefly, the cells were inoculated in a 24-well plate at a density of 2 × 10^4^ cells per well. Following overnight culture to allow for cell adhesion, RhB-LPs were added to different wells and incubated in the dark for durations of 1 h, 4 h, or 12 h. After incubation, the cells were rinsed 2–3 times with PBS, fixed with 4% paraformaldehyde, stained with DAPI to visualize the nuclei, and subsequently observed under appropriate conditions.

### 3.9. In Vitro Cytotoxicity

The cytotoxicity of Lip-*Celastrol*-PROTACs was analyzed by the CCK-8 method employing 4T1 cells. Cells were inoculated into each well of 96-well plates at a density of 5000 cells per well. After the cells had adhered, the original culture medium was aspirated and replaced with fresh culture media containing the drug at varying concentrations (0.5, 1, 1.5, 2, 2.5, or 3 μM). Following incubation periods of 4, 6, 8, 12, and 24 h, each well received an addition of 100 µL of blank medium supplemented with 10% CCK-8 solution. This was followed by a further incubation period of one hour. Subsequently, the plate was transferred to a multimode microplate reader for measurement of the absorbance of the reaction product at a wavelength of 450 nm. The formula utilized for calculating cell viability is as follows:Cell viability (%) = (OD_test_ − OD_blank_)/(OD_control_ − OD_blank_) × 100% (4)
where OD_test_, OD_blank_, and OD_control_ were the optical density values in the test group, blank group, and control group, respectively. As an indicator of in vitro antitumor activity, the half-maximal inhibitory concentration (IC50) was further computed using GraphPad Prism software.

### 3.10. Wound Healing Assay

4T1 cells were seeded in 6-well plates at a density of 9 × 10^5^ cells per well and incubated until complete cell confluence was achieved. A 200 μL pipette tip was then employed to create a cell-free scratch perpendicular to the monolayer. Following this, the cells were washed three times with PBS to eliminate non-adherent cells, thereby ensuring clear visualization of the wound area. Subsequently, the cells were cultured in a medium supplemented with 1% (*v*/*v*) fetal bovine serum containing varying concentrations (100 nM, 200 nM, 300 nM, 400 nM, or 500 nM) of Lip-*Celastrol*-PROTACs for a duration of 24 h. Images were captured using an inverted microscope, and cell migration was assessed and quantified utilizing ImageJ 1.54h software.Migration rate (%) = W_0_ − W_t_/W_0_ × 100%(5)

W_0_ represents the wound area at 0 h, and W_t_ represents the wound area at different time points.

### 3.11. Cell Viability Analysis

To investigate the effect of compounds on cell viability, the live/dead staining assay was applied. Briefly, 4T1 cells were seeded in 6-well plates at a density of 10^5^ cells per well (2 mL per well) and incubated until cell adhesion was established. Following the incubation period, the cells were treated with varying concentrations of Lip-*Celastrol*-PROTACs and further incubated for an additional 24 h. After treatment, residual compounds were removed by washing the cells with phosphate-buffered saline (PBS). The cells were subsequently stained with calcein AM and propidium iodide (PI) fluorescent dyes. To ensure complete removal of excess dye, the cells were rinsed four more times with PBS. Fluorescence images of the stained cells were captured using a fluorescence microscope.

### 3.12. In Vivo Toxicity

The in vivo safety evaluation of Lip-*Celastrol*-PROTACs was studied in female BALB/c mice. Mice were divided into two groups and intravenously injected with stroke-physiological saline solution (SPSS, biosharp BL158A, Shanghai, China) as the control, *Celastrol*-PROTACs, or Lip-*Celastrol*-PROTACs solution. The drug dosage was 100 μL (equivalent to 6 mg/Kg of *Celastrol*-PROTACs), with SPSS as the control. At the end of the experiment, blood samples and major organs (heart, liver, spleen, lungs, and kidneys) were collected from the mice. To assess organ-specific toxicity, hematoxylin and eosin (H&E) staining was performed on major organs. Simultaneously, blood samples were collected for serum biochemical analyses to evaluate toxicity, which included measurements of alanine aminotransferase (ALT), aspartate aminotransferase (AST), creatinine (CRE), and urea (UA).

### 3.13. In Vivo Biodistribution

Female BALB/c mice with established tumors were used as animal models. Briefly, 6-week-old female BALB/c mice were subcutaneously injected with 4T1 cells (1 × 10^6^ cells) suspended in 100 μL PBS to the dorsal flank region. When the tumor volume was about 100 mm^3^, mice were divided into two groups (*n* = 5). DiR, a widely utilized fluorescent dye for in vivo imaging and tracing, was employed as a substitute for *Celastrol*-PROTACs and encapsulated within liposomes (DiR-LPS). Mice were administered either DiR-LPS or free DiR via tail vein injection. At predetermined time intervals ranging from 0 to 120 h post-injection, the mice were deeply anesthetized and subjected to imaging using a small animal imaging system. All animals were euthanized 120 h after injection, at which point images of major organs (heart, liver, spleen, lungs, kidneys) and tumor tissues were collected.

### 3.14. Anti-Tumor Efficacy in Vivo

To establish a 4T1 tumor-bearing mouse model for evaluating the antitumor efficacy of Lip-*Celastrol*-PROTACs in vivo, six-week-old female BALB/c mice were subcutaneously injected with 4T1 cells (1 × 10^6^ cells) suspended in 100 μL of PBS into the dorsal flank region. Once the tumor volume reached approximately 50 mm^3^, the mice were randomly assigned to treatment groups, model groups, and blank control groups, with five mice allocated to each group. The tumor volume and mouse weight were recorded every two days during the treatment period. Tumor volume was calculated using the following formula: Tumor volume = Length × Width^2^/2. At the end of treatment, blood from the mice was collected through eyeball extraction, and serum was separated by centrifuging the blood sample for biochemical analysis. Meanwhile, tumor tissues were collected, photographed, and sliced for H&E staining. Additionally, tumor weights were recorded. Important organs (heart, liver, spleen, lungs and kidneys) were carefully isolated for H&E staining.

### 3.15. Statistical Analysis

The experimental data were analyzed using GraphPad Prism 10.0 software. Student’s *t*-test was employed to compare two independent groups, while one-way analysis of variance (ANOVA) was utilized for comparisons involving multiple groups. Cell viability assays were performed in triplicate under each experimental condition. The IC50 values were determined through nonlinear regression analysis based on the “log (inhibitor) vs. response” model. All statistical results are presented as mean ± SD. A *p* value < 0.05 was considered statistically significant in all analyses.

## Figures and Tables

**Figure 1 pharmaceuticals-18-01381-f001:**
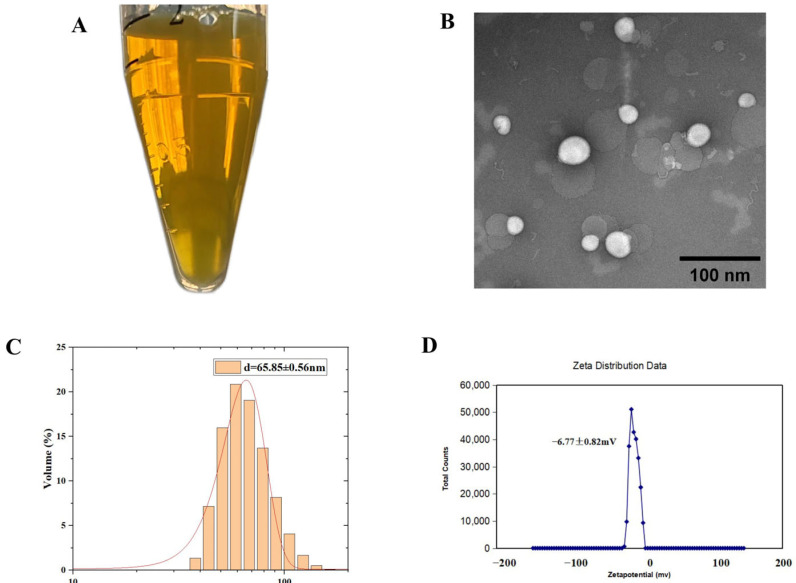
Characterization of Lip-*Celastrol*-PROTACs. (**A**) Representative appearance of Lip-*Celastrol*-PROTACs, (**B**) TEM images of Lip-*Celastrol*-PROTACs, (**C**) particle size distribution, and (**D**) zeta potential.

**Figure 2 pharmaceuticals-18-01381-f002:**
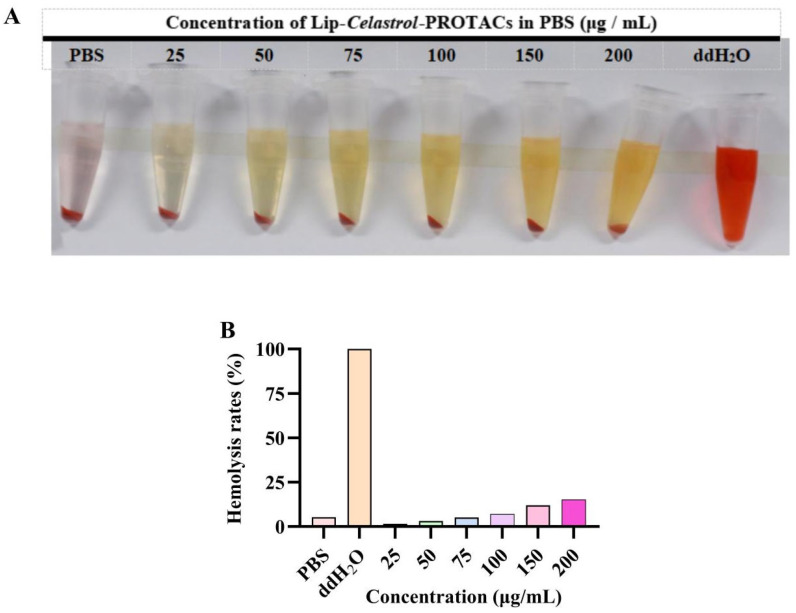
(**A**) Image depicting hemolytic test. Data are presented as mean ± SD from three independent assessments. (**B**) Hemolysis rates.

**Figure 3 pharmaceuticals-18-01381-f003:**
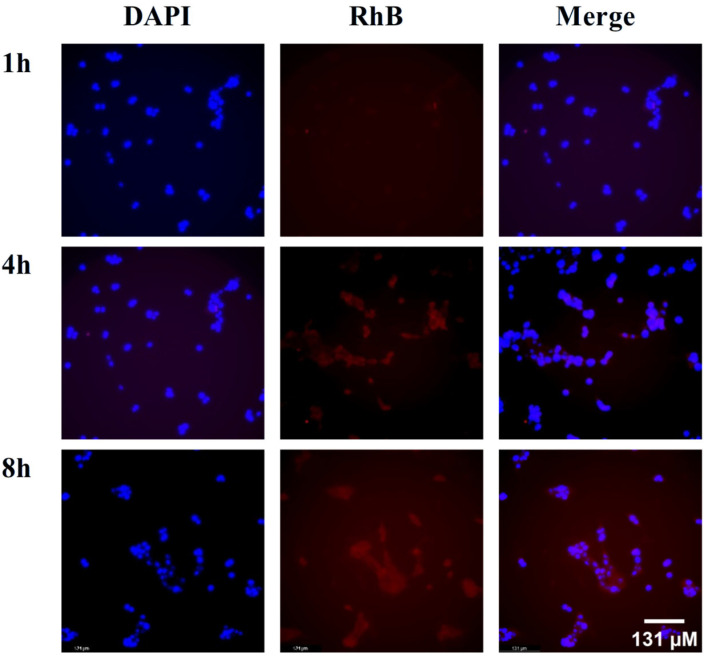
In vitro cell uptake of liposomes against 4T1 cells. Confocal Laser Scanning Microscope (CLMS) image of 4T1 cells incubated with RhB-LPs for 1, 4, and 8 h.

**Figure 4 pharmaceuticals-18-01381-f004:**
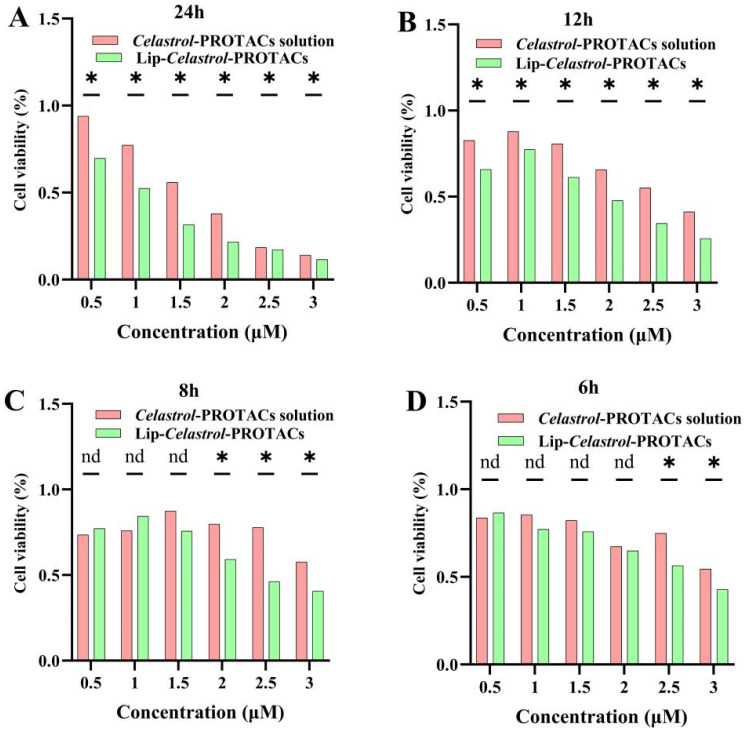
The in vitro cytotoxicity of Lip-*Celastrol*-PROTACs was evaluated using the CCK-8 assay. The cell viability of 4T1 cells incubated for (**A**) 6 h, (**B**) 8 h, (**C**) 12 h, and (**D**) 24 h is presented. Data are expressed as mean ± SD from five independent assessments. nd: *p* > 0.05; *: *p* < 0.05, compared to the *Celastrol*-PROTACs solution group.

**Figure 5 pharmaceuticals-18-01381-f005:**
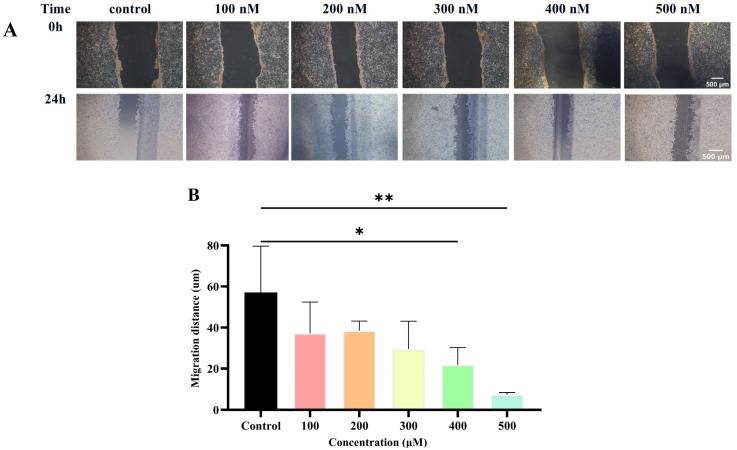
(**A**) Migration images of 4T1 cells at different doses administered. (**B**) Different concentrations of Lip-*Celastrol*-PROTACs inhibited 4T1 cell migration. Data are presented as mean ± SD of five independent assessments. *: *p* < 0.05, **: *p* < 0.01.

**Figure 6 pharmaceuticals-18-01381-f006:**
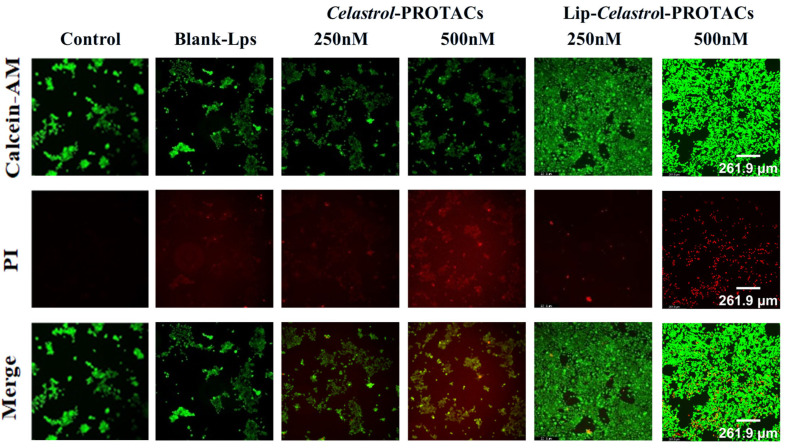
Live cells and dead cells were stained with calcein-AM (green) and PI (red) after incubation of 4T1 cells with different concentrations of *Celastrol*-PROTACs solution and Lip-*Celastrol*-PROTACs.

**Figure 7 pharmaceuticals-18-01381-f007:**
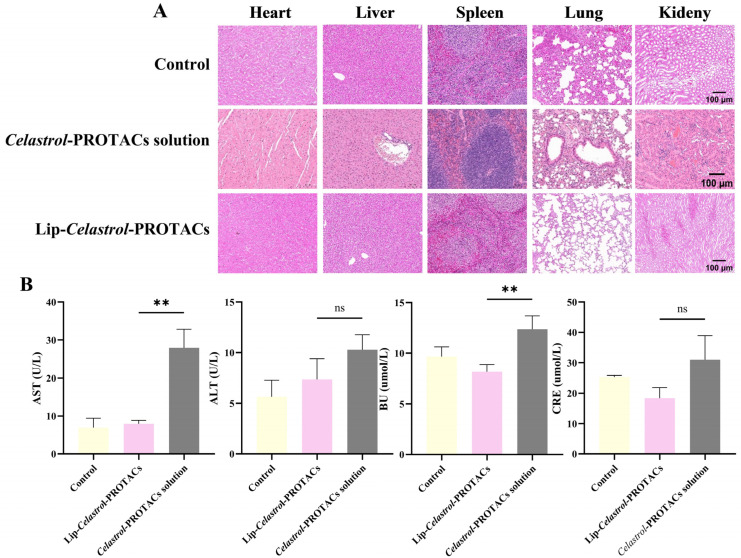
In vivo toxicity of Lip-*Celastrol*-PROTACs on BALB/c mice. (**A**) H&E staining results of main organs after treatment. (**B**) Blood biochemistry data including AST, ALT, CRE, and UA. Data are presented as mean ± SD of five independent assessments. ns: *p* > 0.05, **: *p* < 0.01.

**Figure 8 pharmaceuticals-18-01381-f008:**
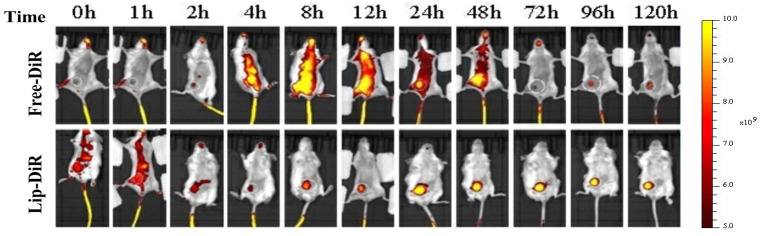
In vivo fluorescence imaging of tumor-bearing mice after intravenous injection with Free-DiR and DiR-loaded liposomes (Lip-DiR) at specific time points.

**Figure 9 pharmaceuticals-18-01381-f009:**
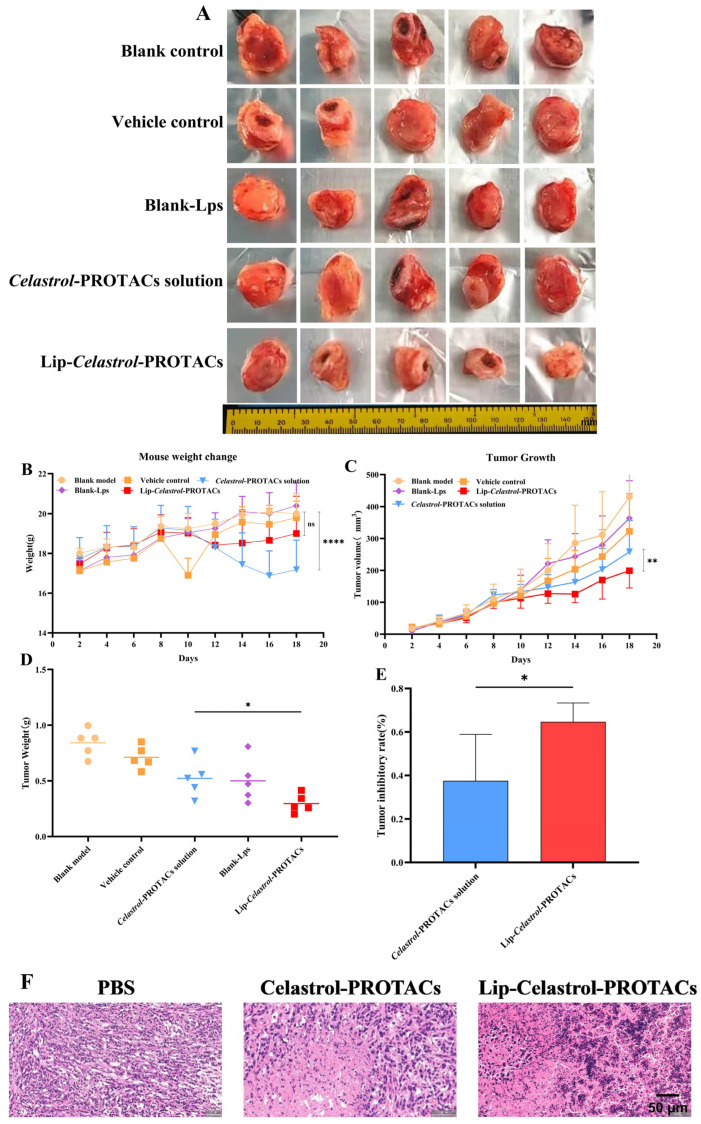
In vivo antitumor effect of Lip-*Celastrol*-PROTACs on BALB/c mice carrying 4T1 tumors. (**A**) Images of tumors, (**B**) body weight changes and (**C**) tumor growth curves of tumor-bearing mice during treatment with different formulations. (**D**) Average tumor weight and (**E**) tumor inhibitory rate harvested from mice at the end of administration. (**F**) Images of 4T1 tumor tissues (H&E staining). Data are presented as mean ± SD of five independent assessments. ns: *p* > 0.05, *: *p* < 0.05, **: *p* < 0.01, ****: *p* < 0.0001.

**Table 1 pharmaceuticals-18-01381-t001:** Encapsulation efficiency and drug loading of Lip-*Celastrol*-PROTACs.

	EE%	DL%
Lip-*Celastrol*-PROTACs	95.73 ± 4.53	10.57 ± 2.96

**Table 2 pharmaceuticals-18-01381-t002:** IC50 (μM) of *Celastrol*-PROTACs solution and Lip-*Celastrol*-PROTACs.

	*Celatrol*-PROTACs Solution	Lip-*Celatrol*-PROTACs
IC50 (μM)	1.770	1.257

## Data Availability

The original contributions presented in this study are included in the article. Further inquiries can be directed to the corresponding authors.

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
