# Peer review of "Development and Evaluation of Liposomal Celastrol-PROTACs for Treating Triple-Negative Breast Cancer"

_pharmaceuticals, 2025, doi:10.3390/ph18091381_

Round 1

Reviewer 1 Report

Comments and Suggestions for Authors

1. In the in vivo toxicity study (Figure 7), Lip-Celastrol-PROTACs showed reduced liver and kidney toxicity compared with free Celastrol-PROTACs. Could the authors provide quantitative histopathological scoring or blinded evaluation results to strengthen the objectivity of these findings?

2. In the in vivo efficacy experiments, the dosing regimen was limited to three intravenous injections (every other day) at 6 mg/kg. Could the authors clarify how this dosage and schedule were selected, and whether alternative dosing frequencies or lower doses were considered to better mimic potential clinical use?

3. The introduction clearly describes TNBC and PROTACs, but the role of Celastrol specifically as a natural product with anti-cancer potential could be elaborated further, including its known pharmacological limitations (e.g., poor solubility, toxicity). This would better contextualize the rationale for formulating liposomes.

4. The discussion of nanotechnology and liposomes could benefit from including more recent references (past 2–3 years) on PEGylated liposomal formulations for TNBC or PROTAC delivery to highlight novelty and connect with current research trends.

5. In the Methods section, the description of the HPLC quantification method (Section 2.2.1) could be more precise if the authors specify how many calibration points were used to construct the standard curve and whether validation parameters (e.g., accuracy, repeatability) were assessed beyond R²

Reviewer 2 Report

Comments and Suggestions for Authors

Review of the Manuscript entitled:

Study of liposomes as drug carriers of Celastrol-PROTACs for treating breast cancer

Dear Editor,

In the present Essay, the authors indicated that based on their previous study, Celastrol-based proteolysis-targeting chimeras (Celastrol-PROTACs) were shown to induce apoptosis in 4T1 cells by selectively degrading GRP94 and CDK1/4 through the endogenous ubiquitin-proteasome system. However, their clinical translation is limited by poor solubility, low targeting efficiency, and liver and kidney toxicity.  To address these limitations, they developed a pegylated liposomal formulation of Celastrol-PROTACs (Lip-Celastrol-PROTACs) and evaluated its therapeutic efficacy and safety profile. Results: Lip-Celastrol-PROTACs significantly enhanced cellular uptake and exhibited potent cytotoxicity against tumor cells. In animal models, Lip-Celastrol-PROTACs demonstrated improved tumor targeting and superior anti-tumor efficacy compared to intravenously administered Celastrol-PROTACs solution. Importantly, Lip-Celastrol-PROTACs markedly reduced the suppressive effect on body weight in mice and alleviated drug-induced hepatotoxicity and nephrotoxicity. Their study demonstrates that pegylated liposomes can enhance the targeting efficiency and reduce the associated toxicity of PROTACs, thereby improving overall therapeutic efficacy. These findings indicate that Lip-Celastrol-PROTACs represent a promising strategy for future clinical applications…

As the topic of this article is vital for treating breast cancer, this article can be acceptable in the journal Pharmaceuticals, but it needs some revisions for improving the quality of the manuscript:

  1. Abstract section was well designed. However, it needs to be improved in content. For example: “Our study demonstrates that pegylated liposomes can enhance the targeting efficiency and reduce the associated toxicity of PROTACs, thereby improving overall therapeutic efficacy.”

The readers wait to see several key data from the results section in the Abstract. Please add some concise and explicit data from reported results.

  1. Furthermore, the title doesn’t show explicitly the applied method in this work. Please improve the Title if possible.
  2. Keywords need to be more remarkable through special words or phrases for example: breast cancer or other professional words.
  3. Page 2, lines 69–70: “Nanotechnology can effectively enhance targeted drug delivery, which not only improves therapeutic efficacy but also mitigates adverse reactions in non-targeted tissues

[16-18].” However, the authors didn’t clarify the drug delivery system and its process  in the method section.

  1. In addition, have the authors encountered drug delivery through modeling and simulation methods? Please clarify it.
  2. Page 3, Schem1; Is the preparation process and therapeutic effect of Lip-Celastrol-PROTACs by authors or literature? Please clarify it. Furthermore, improve its quality especially the texts on the image.
  3. Page 4, lines 140–141 extracted from “2.5.Encapsulation efficiency and drug loading”…To determine the total amount of Celastrol-PROTACs in liposomes, LipCelastrol -PROTACs were diluted 10-fold in methanol to destroy the liposome structure.”

The authors remarked these compounds structures but there is Figure of their structural characterization which is important for understanding the active site of the targeted molecules in encapsulation and drug delivery. It is recommended to add it.

  1. Please improve the quality of Figure 1. It is not clear.
  2. The consequence of the results and discussion section was not clear. Please add a short summary of the mentioned section.
  3. The authors didn’t illustrate the limitation of application of Celastrol-PROTACs (Lip-Celastrol-PROTACs) in vivo. Please explain it.
  4. In addition, the Figures and Schemes in the whole manuscript should be improved with high quality.

Round 2

Reviewer 2 Report

Comments and Suggestions for Authors

Dear Editor,

Regarding the author’s revision, I am pleased to inform my satisfaction of the present form of the manuscript entitled: “Study of liposomes as drug carriers of Celastrol-PROTACs for treating breast cancer” for publication in "Pharmaceuticals".